# Novel Pickering High Internal Phase Emulsion Stabilized by Food Waste-Hen Egg Chalaza

**DOI:** 10.3390/foods10030599

**Published:** 2021-03-12

**Authors:** Lijuan Wang, Jingjing Wang, Anheng Wang

**Affiliations:** 1College of Grain Engineering and Technology, Shenyang Normal University, Shenyang 110034, China; synuwlj@126.com (L.W.); synuwjj@126.com (J.W.); 2Department of Chemistry and Biochemistry, School of Mathematics and Physical Sciences (Chemistry), University of Hull, Cottingham Road, Hull HU67RX, UK

**Keywords:** food waste, high internal phase emulsions, egg, microstructure

## Abstract

A massive amount of chalaza with nearly 400 metric tons is produced annually as waste in the liquid-egg industry. The present study aimed to look for ways to utilize chalaza as a natural emulsifier for high internal phase emulsions (HIPEs) at the optimal production conditions to expand the utilization of such abundant material. To the author’s knowledge, for the first time, we report the usage of hen egg chalaza particles as particulate emulsifiers for Pickering (HIPEs) development. The chalaza particles with partial wettability were fabricated at different pH or ionic strengths by freeze-drying. The surface electricity of the chalaza particles was neutralized when the pH was adjusted to 4, where the chalaza contained a particle size around 1500 nm and held the best capability to stabilize the emulsions. Similarly, the chalaza reaches proper electrical charging (−6 mv) and size (700 nm) after the ionic strength was modified to 0.6 M. Following the characterization of chalaza particles, we successfully generated stable Pickering HIPEs with up to 86% internal phase at proper particle concentrations (0.5–2%). The emulsion contained significant stability against coalescence and flocculation during long term storage due to the electrical hindrance raised by the chalaza particles which absorbed on the oil–water interfaces. Different rheological models were tested on the formed HIPEs, indicating the outstanding stability of such emulsions. Concomitantly, a percolating 3D-network was formed in the Pickering HIPES stabilized by chalaza which provided the emulsions with viscoelastic and self-standing features. Moreover, the current study provides an attractive strategy to convert liquid oils to viscoelastic soft solids without artificial trans fats.

## 1. Introduction

In egg processing plants, chalaza cords are filtered off before the pasteurization step during liquid egg yolk or liquid egg white processing. Over 450 metric tons of chalaza by-products was produced annually after the egg-processing, most of which are disposed. Such abundant resource attracts the attention from researchers to seek for potential utilization for chalaza such as sources for bioactive compounds (Acid, hydrolysates, ovomucin, and lysozyme) [1,2,3]. The protective effects of antioxidant egg-chalaza hydrolysates against chronic alcohol consumption-induced have been demonstrated in mice [4]. However, until now, no works have been carried out about considering chalaza as a valuable source for industrial applications.

Recently, we found that the chalaza’s formation mechanism might show that it is a potential particulate stabilizer, similar with the protein micro-gel [5]. Chalaza contains gelatinous structures appearing at the presumptive sharp and blunt ends along the long axis of the avian egg. At their outer ends, they merge with the outer thick white. Chalaziferous layer touching the vitelline membrane constructs the inner ends [5,6]. The blastoderm is forced into an oblique position, and the embryo will develop perpendicularly to the long axis with its head in the direction of rotation [7]. This unique formation mechanism and the rotation provide chalaza with dual-wettability and insolubility which are crucial features for Pickering emulsifiers [8,9]. Besides, chalaza possesses a lot of advantages to be used as emulsifiers, such as high availability, high nutritional value, outstanding thermodynamic stability, and strong antibacterial properties. Also, chalaza is an egg-originated material that will not be excluded by consumers when it is used in mayonnaise or other food production. Therefore, it is of great value to study this protein particle as an emulsifier to expand the utilization of such material with the abundant source.

Increasing intake of artificial trans fats sourced from lipid products has been recognized as a risk of cardiovascular disease in the medical community [10,11,12]. Briefly, the chance of getting coronary heart disease increase by up to 29% with an increase intaking of 2% in caloric intake from artificial trans fats [13]. Meanwhile, the excessive intake of artificial trans fats was reported to be responsible for causing diabetes and cancers [14]. The dietary intake of trans fats is mainly sourced from partially hydrogenated oils (PHOs) [15]. Recently, the U.S. Food and Drug Administration (FDA) stated to ban the utilization of PHOs in processed food [16]. To meet the new requirements, an effective alternative of PHOs which could mimic the rheology and sensory features of the PHOs are the new challenges for food scientists [17,18,19,20,21]. Oleogelation, which uses the three-phase emulsion system to physically fabricate the artificial butter-like product, is a promising way to meet the increasing demand for healthier food. The formation of oleogels is achieved by trapping the oil drops in the self-assembled gelators to form a 3D-network structure. Unfortunately, most of the gelators reported exhibited improper features such as high cost and low biocompatibility [16,22,23,24]. Therefore, utilizing olegels to secure bioavailable substitutes for PHOs is seriously restricted in the food industry. Another possible routine for developing substitutes for PHOs is the water-in-water emulsion template [24,25,26]. The shortcoming of such a system is the flavor and commercial acceptance. Alternatively, the fabrication of high internal phase emulsions (HIPEs) is a promising strategy to address this challenge via physically transforming liquid oil into solid-like fats without PHOs. This strategy might be achieved by using the full biomass-based chalaza [27,28]. Several food grade materials including gliadin, whey protein microgel, chitosan, cellulose, and zein have been reported to be HIPEs stabilizers. These food biomass-based materials exhibit decent features to be used as Pickering emulsifiers to for the stabilization of HIPEs [29,30,31,32,33,34]. New strategies have also been employed to generate novel Pickering emulsions. Okara dietary fiber particulate obtained via Maillard reaction and protein Hydrolysates showed great efficiency in building the split-new emulsion systems [35,36]. Moreover, the HIPEs stabilized by these particles have also been declared to be a promising template for the constructing of biocompatible scaffold for 3D cell culture. Another feasible routine to utilize the emulsion system is constructing lipid-soluble drug delivery system [37]. The ultra-high oil content in HIPEs system provides itself an excellent drug delivery efficiency. The porosity and biocompatibility provide the food-HIPEs with a promising future. However, most of the raw materials compromise complex preparation progress and need several rounds of modification. A full natural stabilizer that exhibits excellent emulsifying ability is in urgent need.

In this study, we developed a novel approach to fabricate o/w Pickering HIPEs (up to 86% internal phase) using chalaza as a sole particulate emulsifier. The HIPEs stabilized by such particles exhibited features that corresponding to soft solids. The morphology, particle size, and zeta potential of chalaza particles were characterized. Then, chalaza particles were used to develop Pickering HIPEs with different oil fraction. Importantly, the current work opens a hopeful route to create Pickering HIPEs with food-grade status using chalaza as potential delivery carriers of functional ingredients as well as food texture modifiers. In the present study, chalaza was isolated and proved to be eligible to stabilize the HIPEs at different pH values or ionic strengths.

## 2. Material and Method

### 2.1. Material and Chemicals

Hen Eggs and sunflower oil (100% purity) were sourced from Tesco supermarket. Fluorescent dyes, i.e., Nile Blue A, Nile Red, and isothiocyanate (FITC), were sourced from Sigma–Aldrich, Inc. (Shanghai, China). Florisil (60–100 mesh) was obtained from Sigma-Aldrich (Kansas City, MO, USA). MilliQ water was used for all the experiments. All other chemicals were of analytical quality.

### 2.2. Preparation of Chalaza Powders

The eggshell was broken split evenly in half. Then, the whole egg was filtered using a sieve to firstly remove most of the egg white. Subsequently, the integrated chalaza was separated from yolk by tweezers. The chalaza strings were dispersed in 5 mL avian Ringer solution and the dispersion was centrifuged at 1000× *g* for 10 min at 4 °C. The precipitate was washed three times before being re-suspended in Ringer solution followed by centrifugation. The whole procedure was repeated three times and the sediment was collected for freeze-drying. The collected powders were used as chalaza material in the following experiments.

Chalaza suspensions were obtained by adding the different weights of chalaza powder in aqueous solutions. For example, 1 g chalaza powder diluted in 100 mL will be calculated as a 1 wt% chalaza suspension.

### 2.3. Electrophoresis

After the separation of chalaza, it was carefully rinsed three times by ringer solution to remove any substances that come from the egg white. The chalaza was then centrifugated and freeze-dried into powders. We used 0.3 wt% chalaza suspensions for the SDS-Page. Chalaza materials were separated by 7.5% SDS-PAGE (sodium dodecyl sulfate-polyacrylamide gel electrophoresis;) or 3–10% gradient gel “PAGEL” (Atto, Tokyo, Japan). The gels were stained with 0.05% Coomassie Brilliant Blue R-250 (CBB; Nacalai Tesque, Kyoto, Japan). The molecular weight was estimated from a calibration curve obtained by SDS-PAGE molecular weight standards (Bio-Rad, Hercules, CA, USA) or Hi-Mark pre-stained high-molecular protein standards (Invitrogen, Carlsbad, CA, USA).

### 2.4. Particle Size Measurement

Zetasizer Nano ZS (Malvern Instruments, Malvern, UK) and Malvern MasterSizer 3000 were employed to characterize the particle size (Dz). Samples were diluted with purified water to an appropriate particle concentration (e.g., 100 times of dilution, in the present work) before being plotted into the chamber for testing. All measurements were performed at room temperature with three repeats. The results reported are averaged from the three repeats

### 2.5. Optical Microscope Observation

The microstructure of these chalazae at different NaCl concentration values or pH values was firstly analyzed using an optical microscope (OLYMPUS BX51). Samples have different water-immersion objectives. Thirty microliters Chalaza dispersions containing different concentrations of NaCl or at different pH values were shifted and spread onto a glass slide and then covered by glass covers. The observations were carried out at ambient temperature (25 °C).

### 2.6. Emulsion Preparation and Characterization

To generate emulsions, 20 mL of 1.5% chalaza dispersion at different ionic strength (0 M, 0.2 M, 0.4 M, 0.6 M, 0.8 M, and 1.0 M) or pH levels (2, 3, 3.5, 4, 4.5, 5, 6, 8.3) was dispersed into a glass vial. Then different volume of sunflower oil was slowly added to the glass vial where the mixtures were homogenized using Ultra Turrax 25 at 13,500 rpm for 1.5 min. For long term monitoring of the emulsion, the collected emulsions were stored in the fridge set at 5 °C. The microstructure of the freshly formed (1 h) or stored (7 days) emulsions was firstly analyzed using an Olympus microscope (BX51; Leica Microsystems AG, Wetzler, Germany) equipped with different water-immersion objectives. Creaming stability of the emulsions at varying conditions was visually evaluated photographed by Canon 5DII. 10 mL of each emulsion was filled into a glass test tube (1.5 cm internal diameter × 12 cm height) and subsequently stored in a refrigerator (placed in a perpendicular state).

### 2.7. Preparation of HIPEs

HIPEs stabilized by chalaza, formed at different ionic strength (0 M, 0.2 M, 0.4 M, 0.6 M, 0.8 M, 1 M) or pH levels (2, 3, 3.5, 4, 4.5, 5, 6, 8.3), were generated by shearing the mixtures of soy oil and the chalaza suspensions at various ratios of φ (0.75–0.86) and/or particle concentration (0.5–2 wt%) for 120 s using an Ultra Turrax T25 homogenizer (10 mm head) operating at 13,500 RPM. The type (O/W, or W/O) of the resultant HIPEs was evaluated by carrying out the conventional drop test. Briefly, one drop of the emulsion was dispersed in the aqueous phase. Emulsion that could be well dispersed in the aqueous phase was defined as O/W type.

### 2.8. Fluorescence Microscope and Confocal Laser Scanning Microscope Observation

A fluorescence microscope and CLSM were employed to observe the microstructure of the emulsions. Nile red (492/520 nm), The sunflower oil phase was stained with Nile red (492/520 nm), and the chalaza was labeled with FITC (560 nm). The fluorescence microscope was carried out on an Olympus BX51 implemented with a fluorescence burner. Images for confocal observations were obtained with a Zeiss LSM 510 Meta on an Axiovert 200 M microscope (Zeiss, Gottingen, Germany). Regarding high-resolution images, a 63× water-immersion laser technique was used.

### 2.9. Rheology

T HAAKE RS600 rheometer (Thermo Electron Inc., Langenselbold, Germany) equipped with parallel plate geometry (d = 27.83 mm, 1 mm gap) was utilized to measure the rheological measurements of the HIPEs. The viscoelastic parameters were investigated by different frequency sweeps (0.1–10 Hz). Shear rates range from 1 to 50 s−1 were set for flow measurements. All these rheological measurements were performed within the linear viscoelastic region and carried out at 25 °C.

### 2.10. Scanning Electron Microscope Observation

SEM was used to detect the emulsion droplets morphology. One milliliter of the emulsion samples were fixed using an equal volume of 0.5 wt% glutaraldehyde solution. After fixation, the samples were rinsed 3 times with deionized water. The samples were then coated in 10 nm Carbon and imaged using the cryo-scanning electron microscope (Zeiss Evo-60 Cryo-SEM, Jena, Germany Cyro-SEM) system

### 2.11. Statistical Analysis

Three replicates were made for emulsion stability, oil droplet size, protein content, and zeta potential. Confidence profiles of these experiments were conducted on single intervals set at 95% (*p* < 0.05).

## 3. Results and Discussion

### 3.1. Schematic Models of Chalaza Particles

As shown in Figure 1A,B, chalaza existed as a stabilizer for the yolk to help the yolk maintain its steady form. Consumers frequently find chalaza disgusting, and some even consider that chalaza is a bacterium that cannot be eaten. Consequently, the removal of chalaza before egg processing is generally in industrial production, which can be mainly achieved by clipping tools. In this study, we easily removed the chalaza from the egg with tweezers. The chalaza is in the insoluble state and performed like fibers before they are suspended in an aqueous solution (Figure 1C). The collected chalaza was rinsed several times using ringer solution and then freeze-dried into powders. The powders were plotted in the water phase to make chalaza suspension before being used in various experiments (Figure 1D).

### 3.2. Protein Composition of Chalaza

We employed SDS-PAGE to analyze the protein contents in the separated chalaza. The bands were identified according to their molecular weight, which was compared with previous studies [2]. Eight major bands in chalaza were identified (Table 1). Densitometric analysis of the bands enabled us to estimate the relative proportion of ovomucin proteins in the peptide profiles of chalaza (Figure 2 and Table 1). The protein analysis was corresponded to the former works [1,7], indicating that our separation method of chalaza was successful. The principal proteins are ovalbumin (54% of total protein), ovotransferrin (12%), ovomucoid (11%), and lysozyme (3.4%). The protein content of chalaza is similar to egg white. The two fastest migration proteins are α1-and α2 ovomucin, which are considered to be the carbohydrate-poor components of ovomucin. The slow migration one is carbohydrate-rich protein—β ovomucin. Overall, chalaza exhibits a very similar protein composition compared to egg white.

### 3.3. Particle Size, ζ-Potential, and Microstructure of Chalaza Particles

The preparation of the chalaza was conducted using the centrifugation method. The behavior of the collected aqueous dispersions of chalaza at various pH or ionic strengths were discussed to help understand the stability of such particles. Briefly, with increasing ionic strength, the height of sediment in chalaza suspensions slowly increased (Figure 3A) at ionic strength lower than 0.6 M, which could be attributed to the result of the gaining insolubility. Correspondingly, the ζ-potential of chalaza soared from −12 mV to −5 mV with low amount of salt addition (<0.6 M). A steady increase with further salt addition could be observed (Figure 3B).

The particle size characterization was then conducted. The results (Figure 3C) indicated that the size of the particles reached a peak of 700 nm with 0.6 M salt addition where the electricity of particles were mostly neutralized. The results of particle size corresponded to the results in ζ-potential section. To further investigate the performance of the chalaza particles, the microscope observation was carried out. The results (Appendix A) clearly indicated that chalaza particles do not remain uniform at various ionic strength. Plots like particles could be detected at low (<0.2 M) or high (>0.6 M) ionic strength. However, with medium salt addition (between 0.2 M and 0.6 M), the particles performed more like curled clusters, which could show that the salt addition changed the surface electricity of chalaza and thus, altering their size and shape (Figure 3C). Similar set of characterizations were also carried out to chalaza at different pH levels.

The suspensions of the chalaza at different pH levels were concerning similar turbidity and sediment height (Figure 4A). The ζ-potentials of the aqueous dispersion obtained from Zeta Nano ZS were set as a function of the pH (Figure 4B). When the pH was reduced to 2.0, the particles became highly positively charged (z > +13 mV). The electricity of chalaza was nearly neutralized when the pH was increased close to 4.0. With further pH increasing, the particles could contain strong negative electricity when the pH is higher than 5 (z < −12 mV).

The particle size of the chalaza (Figure 4C) at any pH level conformed to a narrow range. At rather low or high pH levels, the size of the chalaza was of the smallest, around 300 nm. The strong repulsion force sourced from high surface charging at pH far away from the isoelectric point significantly reduce the particle aggregation. When the pH was close to the isoelectric point of chalaza (pH 4), where the electricity was neutralized and the electrostatic repulsion was inhibited, the particle became relatively unstable and aggregated severely.

### 3.4. Emulsion Characterization

#### 3.4.1. Effect of pH and Particle Concentration

Surface charging and colloidal stability were strongly affected by pH levels. In the present study, highly positive surface charging and negative charging were detected at pH < pI and pH > pI, respectively (Figure 4B). With pH close the isoelectric point of chalaza particles, the electrical repulsion was neutralized. The chalaza lost its hydrophilicity nature and aggregated severely (Figure 4). Accordingly, a constant chalaza concentration of 1.5 wt% was set to investigate the emulsions’ stability against coalescence and flocculation at different pH levels. Typically, the pH of the aqueous Chalaza-containing phase varied, and all values (including 2.0, 3.0, 3.5, 4.0, 4.5, 5, 6, and 8.3) were selected to be sufficiently far from the isoelectric point of chalaza (pI 4). As a result, chalaza particles from strongly positively charged to negatively charged could be utilized to stabilize oil-in-water emulsions. Additionally, similar experiments were also conducted to chalaza with different salt addition. Particle concentration has a profound effect on emulsion stability against coalescence as well as on the droplet size of the HIPEs. To firstly select an adequate particle concentration to stabilize the HIPEs, four different sets of 5 mL chalaza suspensions were prepared (0.5 wt%, 1.0 wt%, 1.5 wt%, and 2.0 wt%) at pH 4 and then mixed with 15 mL sunflower oil to generate 75% oil phase. The collected HIPEs emul-gels were plotted into oil or water to check whether they were O/W type emulsions (Figure 5B,C). The emulsions which could be diluted and dispersed in water were O/W type emulsion. A long term storage of the HIPEs witnesses no water phase at the bottom after 7 days when the particle size exceeded 1 wt% (Figure 5A), demonstrating that the chalaza particles were sufficient to stabilize emulsions with 75% oil phase beyond such concentration. Consequently, all the following experiments were conducted with 1.0 wt% chalaza suspensions.

#### 3.4.2. Development and Appearance of Pickering HIPEs

Using a T10 basic Ultra Turrax homogenizer, the mixtures containing sunflower oil and the chalaza aqueous dispersion was sheared to produce the Pickering HIPES. The appearance of Pickering HIPEs after 2 h and 15 days as a function of pH levels could be seen in Figure 6A,B. In these series, 75% internal phase(dispersed phase) was set as a fixed value (Figure 6A). To satisfy the increasing demand on healthier food, surfactant-free emulsion-based food attracts more researchers working on it. In the emulsions with internal phase fractions of 75% are all surfactant-free o/w Pickering HIPEs. In contrast, HIPEs stabilized by surfactants are usually w/o [26] and extensive surfactants (5–50%) are frequently demanded to effectively produce stable HIPEs [20]. Our emulsions did not undergo creaming of dispersed phase fractions at pH around the isoelectric point (3, 3.5, and 4) after 15 days’ storage. For these three pH levels, the stability of emulsions could overcome coalescence and creaming, further revealing that chalaza is a potential Pickering stabilizer for HIPEs. The novel chalaza particles could attach and anchored on the interfacial layer, compressing deformed droplets to densely pack into a 3D percolating network. The network enabled the emulsions to remain stable against creaming and coalescence with oil phase lower than 85% at pH 3 and 3.5 (Figure 6C). Unfortunately, the emulsion system would collapse with internal phase volume factions higher than 86%, except the set of pH 3 (Figure 6D). The microscopic observation could demonstrate that the HIPEs formed at pH 3 could remain stable after days of storage (Figure 6E). Additionally, the full sets of experiments were also done to chalaza at different ionic strengths. The HIPEs with an oil fraction of 75% could only be formed at an ionic strength of 0.6 M, which was consistent with the highest particle size in our former experiment (Appendix A). Besides, the percolating net-work in HIPEs also ensured the emulsions’ stability against creaming with oil content lower than 85% with 0.6 salt addition (Appendix A).

Briefly, we developed a novel edible Pickering HIPEs stabilized by chalaza particles which contains internal volume fraction up to 86%. Such high internal fraction provide itself with the possibility to be labelled as a PHO alternative. The volume fraction of the chalaza dispersion ranged from 0.1 wt% to 2 wt%. Besides, we also characterized the size distributions of Pickering HIPEs. The HIPEs showed a nearly monodispersed droplet size distribution, and the maxima of the profiles shifted towards a lower size when particle concentrations increase. The mean particle size quantitatively reflect the size of Pickering emulsions. Besides, the D4,3 of Pickering HIPEs increased gradually from 20 ± 0.5 μm to 145.1 ± 0.5 upon increasing salt addition before the ionic strength reached 0.6 M, which could confirm that the more concentrated the initial dispersion was, the bigger the drops were (Figure 7A). Differently, with increase in pH levels before reaching the isoelectric point, the size of emulsions decreased from 54 ± 0.5 μm to 32 ± 0.5 μm. Then, the emulsion size dramatically increased to 59 ± 0.5 μm (Figure 7B). The result combined with our observation in the appearance of HIPEs could reveal that the emulsions were in the most stable state around the isoelectric point of chalaza. This would help to industrially produce mayonnaise-like emulsions using chalaza since the commercially used mayonnaise were commonly of acidic pH [38,39]. Besides, the chalaza transformed the liquid sunflower oil into a solid state by stabilizing the three phases (W/O/particles) into HIPEs, which contained self-standing feature after the cups were inverted. In principle, the droplet-droplet interaction through the emul-gel in Pickering HIPEs formed a percolating network architecture and provided additional stabilization against coalescence.

#### 3.4.3. Centrifugation Stability

Emulsion creaming would be significantly accelerated using centrifugation since the water was drained from the Pickering emulsions [40,41] (Figure 7C,D and Appendix A). To characterize their stability, the CI value of the emulsions after the centrifugation was recorded. HIPEs stabilized by chalaza suspension at all pH levels remained stable after severe centrifugation. The CI values were decreased from 51.5% to 21.5%, 23.5%, and 29.5% when the pH levels were increased from 2 to 3, 4, and 5 respectively. The CI values slowly decreased after the pH exceeded the isoelectric point. Similarly, the HIPEs stabilized by chalaza at different ionic strengths only remained stable with 0.6 M salt addition. The creaming index slightly decreased from 51.5% to 26.5%. The above results indicated that the water (continuous phase) that existed in the 3D emulsion network was drained out from the Pickering HIPEs during centrifugation. No oil emersion was detected in both Pickering emulsions and HIPEs after centrifugation, indicating that the emulsions were resistant to the high stress of centrifugation. Chalaza at adequate pH levels or ionic strength were proved to be resistant to coalescence during the centrifugation.

#### 3.4.4. Microscopic Observation of HIPEs

Storage stability is a key indicator of any formula because it determines to some degree whether or not a product is suitable for the intended use [42,43]. This could be also be shown from the microscopic observations (Figure 8 and Figure 9). Stable HIPEs with 75% or 80% oil could be formed at pH levels range from 2–8.3, indicating the strong emulsifying nature of the chalaza material. The fluoresce microscope (Figure 9) demonstrates that the chalaza particle may worked as a particle stabilizer anchoring on the emulsion droplet to fight against coalescence. With increasing oil fraction, only at pH levels (3, 3.5, 4, 4.5) around isoelectric point, uniform stable emulsions could be observed. pH levels of 3 and 3.5 are the two ideal conditions for fabricating chalaza stabilized-HIPEs with 85% oil. These results are in correspondence to our result obtained from size distribution and appearance observation confirming the effectiveness of chalaza as a solid emulsifier to stabilize Pickering HIPEs.

#### 3.4.5. CLSM

The microstructure of the emulsion including interfacial structure and coalescence state of droplets could be detected using CLSM, which were key features related to the physical performance of the Pickering HIPEs [43,44]. Nile Red (green) and FITC (red) were used to stain sunflower oil and chalaza, respectively. Figure 10 shows the CLSM micrographs of representative Pickering HIPEs samples in overlap (C and D) field. In brief, green fluorescence resided in spherical droplets while the red fluorescence emerged mainly at the perimeter of the droplets in Pickering HIPEs, further demonstrating that the formed Pickering HIPEs were o/w emulsions where the chalaza interface layer appears around dispersed droplets. The previously reported protein or polysaccharide particles stabilized HIPEs exhibited similar interfacial behavior when the particles were anchored [44]. The particles would form a fort against coalescence on the droplets outer layer and the excessive particles may existed in the continuous phase to work as a network helping the emulsion stable. In combination with these results and microscopic observation (Figure 9A,B), we propose that the chalaza particles existed as particle hindrance between droplets to prevent coalescence. CLSM and Microscopic observation trials also reflected that the adjacent droplets were closely packed with each other to form percolating network structure which endowed the Pickering HIPEs viscoelastic solid-like feature and stability against creaming and coalescence. These results could show that the isolated chalaza particle could remain in its original form after attaching on the interface, behaving like a solid barrier to avoid droplets flocculation.

#### 3.4.6. Scanning Electron Microscope Observation

To address the morphology of the emulsion droplets. The cryo-SEM was employed after sample mixing. As shown in Figure 11, at high oil content, the droplets remained uniform in structure and size. The protein particles could be observed packaging around the oil droplets in the Figure 10B. This indicates the obvious Pickering stabilization mechanism of the chalaza particles. More importantly, the excess of the particles was observed in the continuous phase, acting as a hindrance against the coalescence and forming a percolating network.

#### 3.4.7. Rheological Property of Pickering HIPEs

Besides the size distribution and microstructure, the rheological property is also treated as a major role in the performance of the emulsions, e.g., appearance and stability. We characterized the amplitude sweep profiles of Pickering HIPEs (Figure 12). Similar amplitude sweep profiles were observed for Pickering HIPEs at pH4 (Figure 12B,D) with different oil content. With increasing oil content from 75% to 85%, both of the G’ and G’’ increased, which could demonstrate the increasing viscosity brought by oil addition. For all of the samples, G’’ was always slightly lower than G’ at relatively lower amplitudes, the HIPEs could still maintain the stability at 100 Pa which strongly indicates its excellent rheology performance. For the HIPEs stabilized by chalaza particle at 0.6 M with different oil addition, high oil content contained weaker performance in both of the modules. Further increasing amplitudes could a cause distinct fall (after 55 Pa), revealing the yielding of the structure at higher stress, which could be attributed to the collapse of the structure at high stress.

Pickering HIPEs were subjected to fixed stress (1 Pa) to study their response to frequency sweep in the linear response region. As expected, the corresponding G’’ was always lower than G’ throughout the frequency range at pH4 (Figure 12D). Rheological data corresponded to the results in visual appearance and storage stability which indicated the viscoelastic and self-supporting features of HIPEs. Interestingly, the G’’ and G’ were not reduced at high oil content. The interfacial behavior of the chalaza particles played a crucial role in altering the viscoelastic property of Pickering HIPEs. The chalaza-coated droplets in HIPEs system contributed to the stiffness of formed solid-like materials. The G’’ and G’ are not consistent with 0.6 M salt addition at low oil content (75%). Both modules were bumping severely with the frequency increasing but were still stable overall. The Pickering HIPEs stabilized by chalaza possess a promising potential to be used as a substitute for PHOs, which was thought to increase the risk of cardiovascular diseases.

## 4. Conclusions

To conclude, we report firstly the usage of chalaza particles, arepresentative of natural egg waste materials, as a potential Pickering particulate emulsifier for HIPEs stabilization. Using a facile route, the sub-micron chalaza with the suitable physical–chemical performance were generated at different pH or ionic strength. We successfully prepared stable Pickering HIPEs with internal phases of up to 86% with low particle concentrations (0.5 wt%). The chalaza particles were effectively adsorbed and anchored at the oil–water interface, best at pH 3 and ionic strength of 0.6 M, exerting electrical hindrance between the droplets. Concomitantly, the compressed droplets in Pickering HIPEs formed a percolating 3D-network framework endowing the emulsions viscoelastic and self-supporting features. The rheology performance of the chalaza stabilized HIPEs shows that this strategy could potentially transform liquid oils into viscoelastic solid-like gels without artificial trans fats. Besides, these HIPEs open a new window for the utilization of such abundant food waste material. Moreover, the current work is conducive to the construction of bio-available chalaza base-emulsion formulation with widespread applications in cosmetics, food, and pharmaceuticals.

## Figures and Tables

**Figure 1 foods-10-00599-f001:**
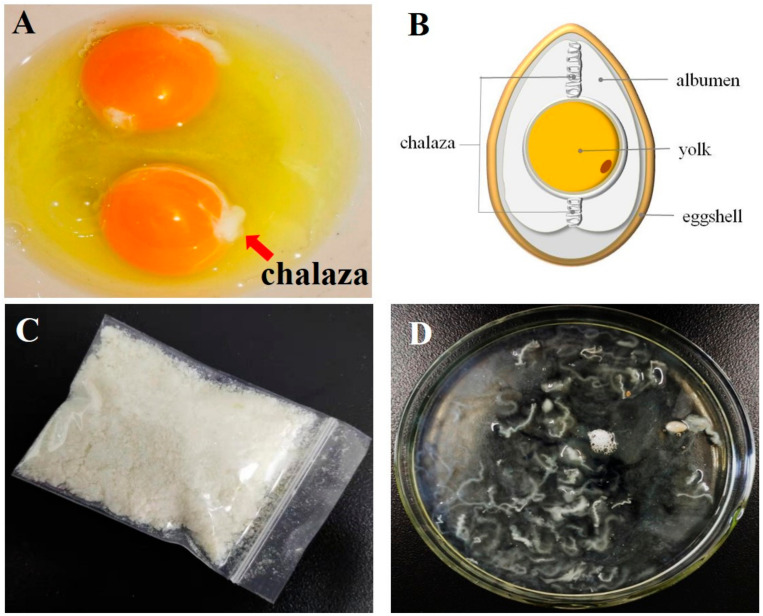
Chalaza at different stages: (**A**) Un-separated chalaza in the hen egg. (**B**) Schematic model of the chalaza in fresh egg. (**C**) Dry powder of chalaza obtained by freeze-drying. (**D**) Separated fresh chalaza samples from hen egg.

**Figure 2 foods-10-00599-f002:**
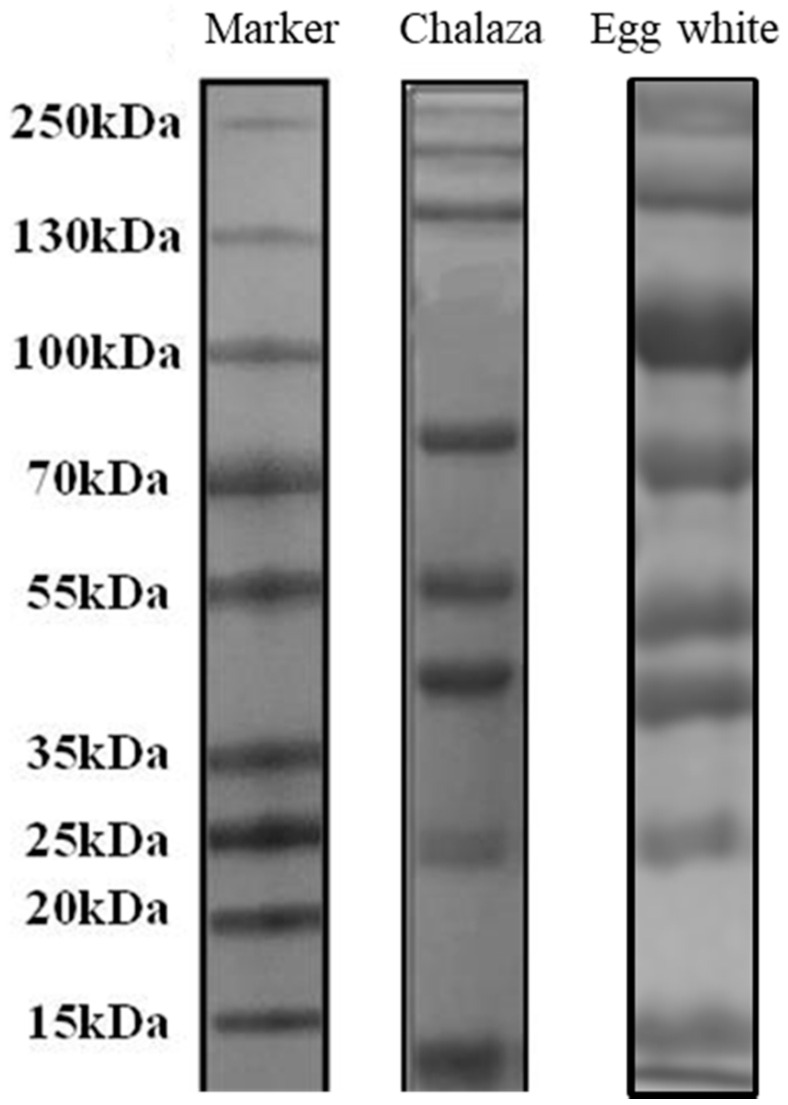
Protein profile of chalaza obtained from SDS-page analysis.

**Figure 3 foods-10-00599-f003:**
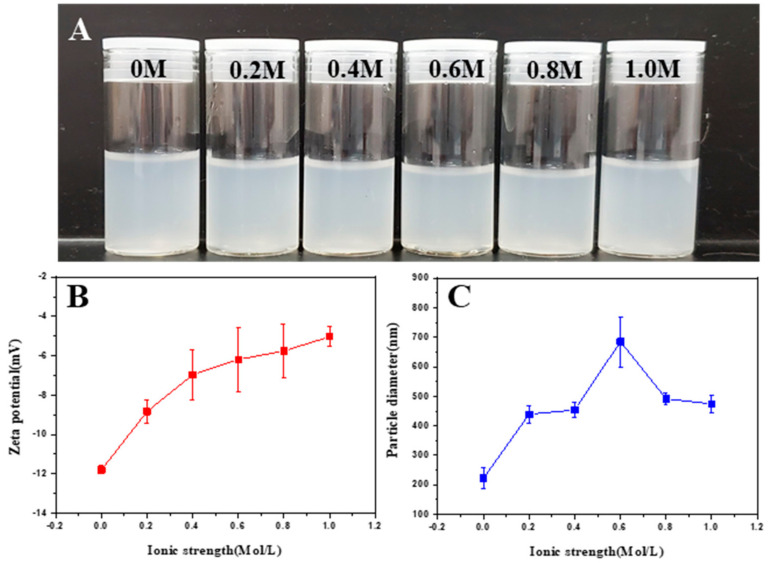
Characterizations of chalaza suspensions at pH 6.5: (**A**) Appearance of chalaza suspension at different ionic strength. (**B**) ζ-potential of chalaza suspension at different ionic strength. (**C**) Particle size of chalaza at a different ionic strength.

**Figure 4 foods-10-00599-f004:**
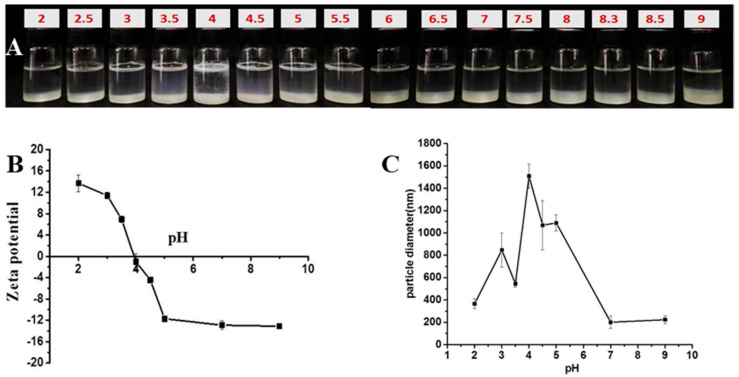
Characterizations of chalaza suspensions: (**A**) Appearance of chalaza suspension at different pH levels. (**B**) ζ-potential of chalaza suspension at different pH levels. (**C**) Particle size of chalaza at different pH levels.

**Figure 5 foods-10-00599-f005:**
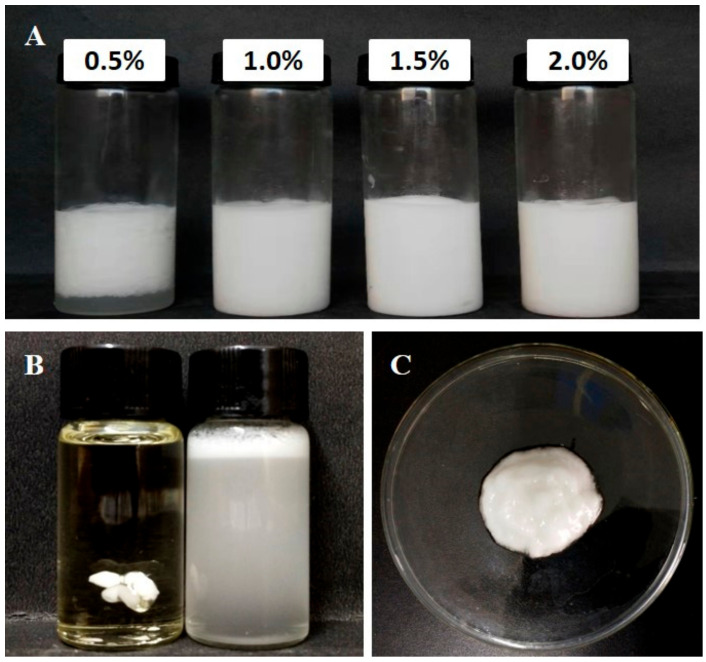
Appearance of chalaza stabilized high internal phase emulsions (HIPEs) with 75% oil: (**A**) HIPES stabilized by different chalaza particle concentrations. (**B**) Comparing observations on the O/W or W/O type of the 75% HIPEs stabilized by 1.0 wt% chalaza particles. (**C**) Appearance of the formed HIPEs emul-gel with 75% oil stabilized by 1.0 wt% chalaza particles.

**Figure 6 foods-10-00599-f006:**
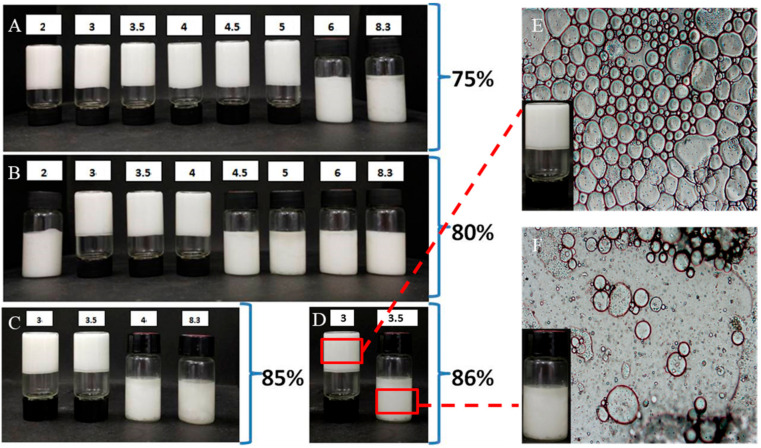
Appearance of HIPEs stabilized by 1.5%wt chalaza suspension at different pH levels with various oil volume: (**A**) 75%, (**B**) 80%, (**C**) 85%, (**D**) 86%, and (**E**) Microscopic observation of 86% HIPES stabilized by 1.5%wt chalaza at pH 3. (**F**) Microscopic observation of HIPES stabilized by 1.5%wt chalaza at pH 3.5 with 86% oil content showing the phase inversion of the system.

**Figure 7 foods-10-00599-f007:**
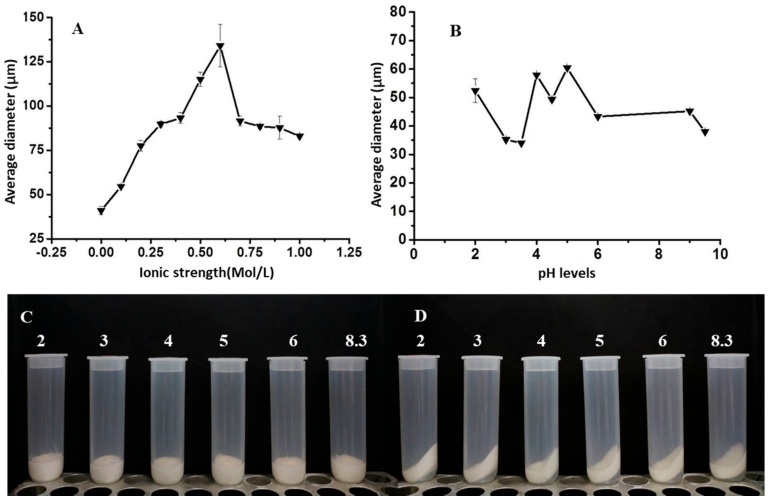
Characterization of HIPEs stabilized by 1.5%wt chalaza with 75% oil at different conditions: (**A**) Average diameter of emulsion droplets at different ionic strength at pH 6.5. (**B**) Average diameter of emulsion droplets at different pH levels without salt addition. (**C**) Appearance of the HIPES’ stability at different pH levels without salt addition before centrifugation. (**D**) Appearance of the HIPES’ stability at different pH levels without salt addition after centrifugation.

**Figure 8 foods-10-00599-f008:**
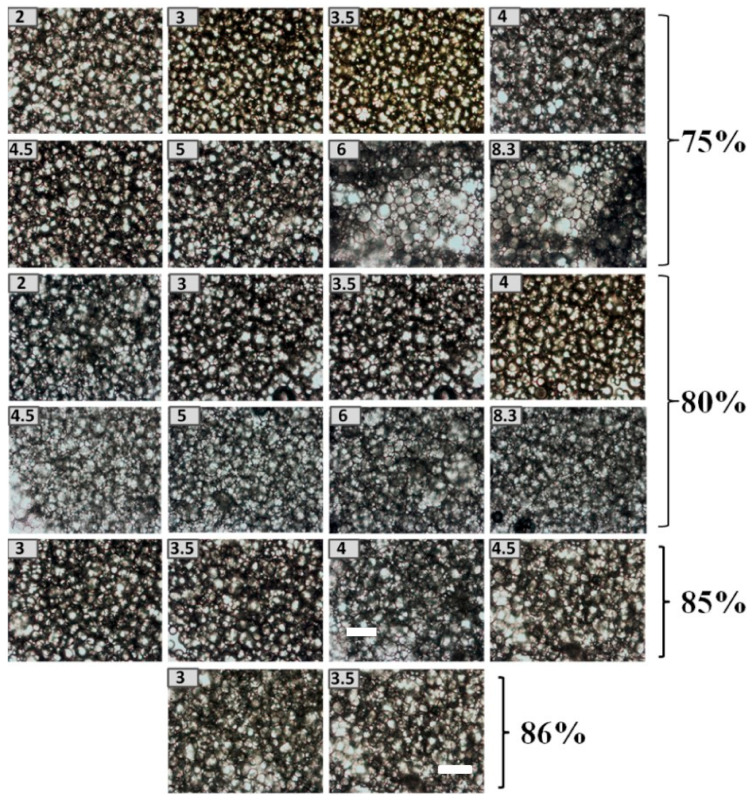
Microscopic observations of HIPEs stabilized by chalaza with the different oil phase. The bar is 50 μm.

**Figure 9 foods-10-00599-f009:**
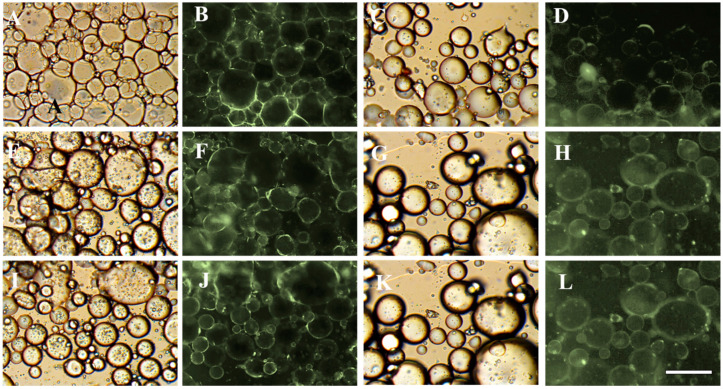
Bright field and fluorescence microscopic observations of HIPEs stabilized by chalaza with different oil phase ((**A**–**D**): 75%) ((**E**–**H**): 80%) ((**I**–**L**): 85%) at pH 6.5 with 0.6 salt addition (**C**,**D**,**G**,**H**,**J**,**K**) or pH 3.5 without salt addition(**A**,**B**,**E**,**F**,**H**,**I**). The bar is 100 mm.

**Figure 10 foods-10-00599-f010:**
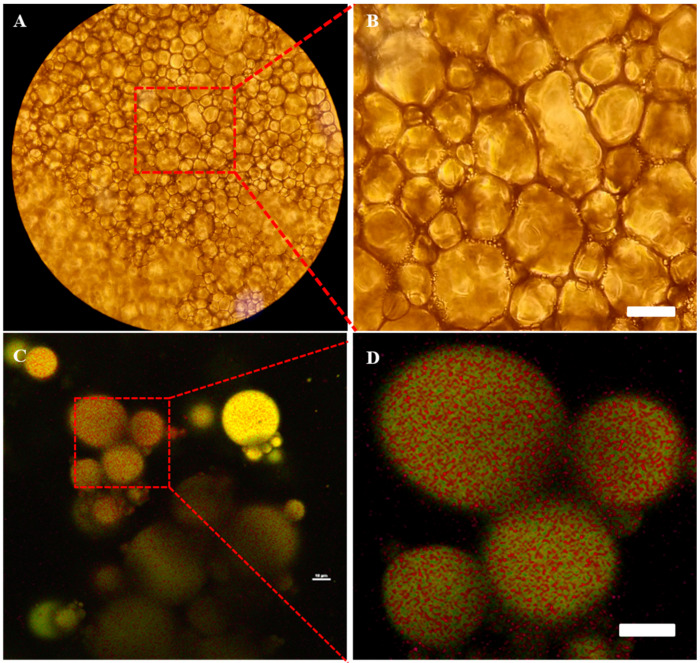
Microscopic observations (**A**,**B**) and CLSM (**C**,**D**) of HIPEs stabilized by chalaza at pH 4 with 80% oil volume. The bar is 50 μm.

**Figure 11 foods-10-00599-f011:**
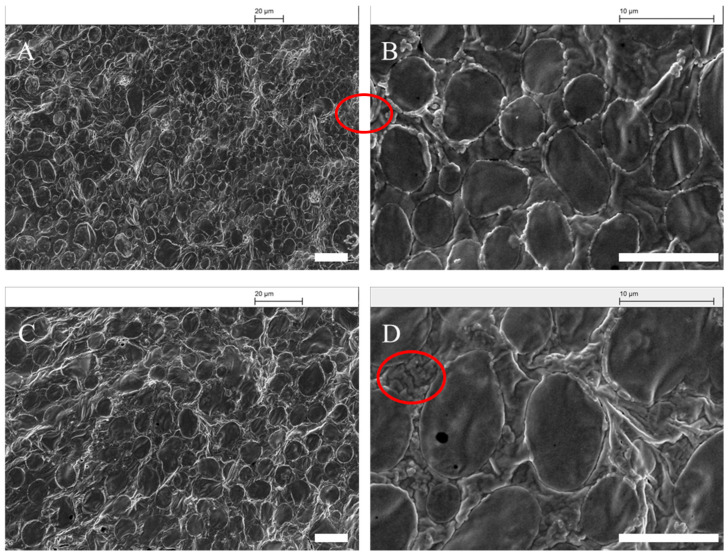
SEM image of emulsion droplets at different oil contents: (**A**,**B**) 75%, (**C**,**D**) 80%. The bar is 20 mm in (**A**,**C**), 10 mm in(**B**,**D**).

**Figure 12 foods-10-00599-f012:**
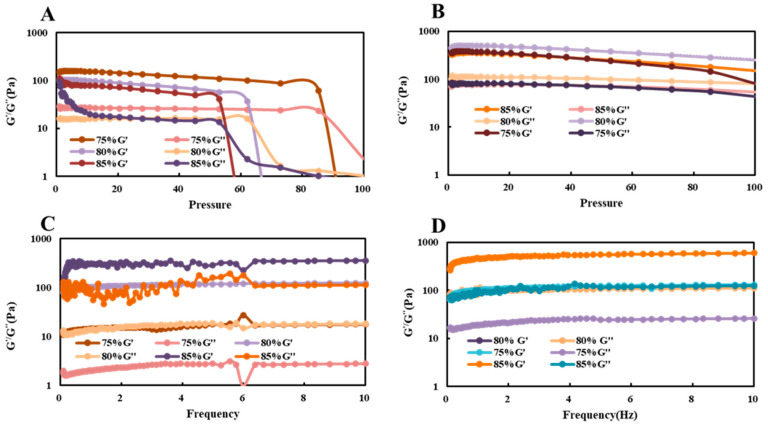
Evolution of storage (G’) and loss (G’’) moduli with the stress (**A**,**B**) and frequency (**C**,**D**) sweeps for the Pickering HIPEs with different oil content with 0.6 M salt addition (**B**,**D**) or at pH3.5 (**A**,**C**).

**Table 1 foods-10-00599-t001:** Protein profile of hen egg chalaza obtained from the SDS-PAGE.

Molecular Weight (kDa)	Identification
400	β-ovomucin
220	α2-ovomucin
150	α1-ovomucin
78–80	ovotransferrin
43–45	ovalbumin
23	ovomucoid
14.4	lysozyme

## Data Availability

Not applicable.

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
