# Peer review of "Novel Pickering High Internal Phase Emulsion Stabilized by Food Waste-Hen Egg Chalaza"

_foods, 2021, doi:10.3390/foods10030599_

Round 1
Reviewer 1 Report
The overall quality of the manuscript is good.
Title clearly describes what the manuscript is about. Abstract adequately describes the study, principle results and conclusions. Employed experimental methods are adequate, sufficiently clear and complete to allow repetition of the work. Data are properly analyzed and interpreted to support the conclusions. Pictures and tables are satisfactory and interpreted correctly. Relevant issues in discussion are adequately discussed. Cited references are appropriate.
This is an interesting manuscript which is a continuation of equally interesting research of the scientific team. The rheology performance of the chalaza stabilized HIPEs shows that this strategy could potentially transform liquid oils into viscoelastic solid-like gels without artificial trans fats. Besides, these HIPEs opens a new window for the utilization of abundant food waste material, hen egg chalaza. Moreover, the current work is also conducive to the construction of bio-available chalaza base-emulsion formulation with widespread applications in cosmetics, food and pharmaceuticals.
Further observations and methodological assumptions testify to the high professionalism and scientific curiosity of the authors. The results and their presentation are complement of many previous research.
Author Response
We really appreciate the reviewer’s hard work on looking through our paper. The reviewers‘ comments are very helpful in our following experiments. We really thank the comments from this reviewer! We will keep revising the paper until it reaches the standard of Foods.
Reviewer 2 Report
It is worth exploring the possibility to utilize egg chalaza, an industrial waste, as a Pickering material. However, there are so many mistakes in the format (style). Please check the manuscript very carefully to correct them. I will show you some of them. Also, I have some comments on the content.
- Format (style)
・space between number and unit, comma and number
・Fig. or Figure?
・G’ or G´?
Line 35: Check the periods.
Line 125: Delete this line.
Line 147: Describe the citation.
Line 149: “analyze” is wrong.
Line 151-152: This sentence is grammatically incorrect.
Line 172: Check the periods.
Line 274: Add “%” after “26.5”.
Figure 8, 9, 10, and 11: Add a period at the end of a sentence.
Figure 9; Nacl is misspelled.
- Content
Figure 2: The bands are so thick and not clear, and the shape of the band under 15kDa is round. Also, the molecular weight of OVA should be around 45,000 Da, but the corresponding band is located over 55kDa. Check the data carefully.
Figure 3: What is the pH of the suspensions?
Figure 4: Explain why the particle diameter is not the maximum at pH 3.5? Based on the result of the zeta potential, the repulsion force should be minimum at pH 3.5.
Line 350-352: I do not understand how it can be used as a substitute for margarine.
Author Response
Reviewers two
It is worth exploring the possibility to utilize egg chalaza, an industrial waste, as a Pickering material. However, there are so many mistakes in the format (style). Please check the manuscript very carefully to correct them. I will show you some of them. Also, I have some comments on the content.
We really appreciate the suggestion from the reviewer. We have carefully revised the manuscript based on your advice.
- Format (style)
・space between number and unit, comma and number
・Fig. or Figure?
・G’ or G´?
Line 35: Check the periods.
Line 125: Delete this line.
Line 147: Describe the citation.
Line 149: “analyze” is wrong.
We apologize for these format fault. We have carefully checked and revised these faults.
Line 151-152: This sentence is grammatically incorrect.
We thank the reviewer for this suggestion, this sentence has been rewritten. The text goes as follow:The protein analysis was corresponded to the former works[1,7], indicating that our separation method of chalaza was successful. The principal proteins are ovalbumin (54% of total protein), ovotransferrin (12%), ovomucoid (11%), and lysozyme (3.4%). The protein content of chalaza is similar to egg white. The two fastest migration proteins are α1-and α2 ovomucin, which are considered to be the carbohydrate-poor components of ovomucin.
Line 172: Check the periods.
Line 274: Add “%” after “26.5”.
Figure 8, 9, 10, and 11: Add a period at the end of a sentence.
Figure 9; Nacl is misspelled.
We apologize for these format fault. We have carefully checked and revised these faults.
- Content
Figure 2: The bands are so thick and not clear, and the shape of the band under 15kDa is round. Also, the molecular weight of OVA should be around 45,000 Da, but the corresponding band is located over 55kDa. Check the data carefully.
1) We thank the reviewer for this comment. We have reconducted the Protein analysis of the last band. The quality has been improved. The ovalbumin is located between the 35kda and 45kda and we believe this is clearly shown in our result. The protein located at 55kda is a subunit of ovotransferrin.
Figure 3: What is the pH of the suspensions?
2) We apologize for missing this important data. The pH is 6.5 and we have added this to the figure caption.
Figure 4: Explain why the particle diameter is not the maximum at pH 3.5? Based on the result of the zeta potential, the repulsion force should be minimum at pH 3.5.
3) We appreciate this suggestion from the reviewer. We have re-tested the samples for Size and Zeta-potential tests. The isoelcetric point is 4. And the maximum size is also reached at 4. We have changed the figure.
Line 350-352: I do not understand how it can be used as a substitute for margarine.
4) We are grateful to this advice. The rheology property of this Chalaza stabilized HIPEs make it possible to be a promising substitute of the PHOs. It contains a very similar property to the margarines.

Reviewer 3 Report
The work by Lijuan Wang et al. is an interesting one. It has the scientific quality and novelty. However, the following are my concerns:
-The abstract requires revision. Authors should make their results more pronounced in the abstract.
-The rendition in lines 25-27: “Especially the protective effects of antioxidant egg-chalaza hydrolysates against chronic alcohol consumption-induced have been demonstrated liver steatosis in mice [4]”is poor. Revise and rewrite.
-The statement in line 29 needs a reference.
-Line 30 also has poor construction. Revise and rewrite.
-Line 44: Is that a reference? If so, it should be according to the journal’s format.
-Lines 53-54 rendition is poor. Revise and rewrite.
-The introductory section should provide more on state of the art of emulsions, types etc. These recent references are recommended: https://doi.org/10.3390/foods9020143; https://doi.org/10.3390/foods9050636; and https://doi.org/10.1016/j.lwt.2021.111024
Authors should try to improve the whole section too.
-Authors must provide the location name in the materials/methods section. Local market, where?
-Line 80: “…were used for...”
-Line 147. You must provide reference(s).
-Lines 149-155: The renditions are poor. Revise and rewrite.
-What is the difference between sections 3.1 and 3.3 with respect to their titles?
-Moreover, section 3.3 is poorly written and discussed, lacking flow and harmony. Authors failed to explain the importance of ionic concentration and pH levels in chalaza suspensions. Then, substantial references were lacking.
-Please, note that the whole of section 3.4 lack proper discussion, grammatical accuracy and adequate references!
-Authors should ensure a constant flow between the abstract the main text.
-My final concern is that general grammatical errors and poorly written sentences could be found from the abstract to the acknowledgements. A CAREFUL revision is required.
Author Response
The work by Lijuan Wang et al. is an interesting one. It has the scientific quality and novelty. However, the following are my concerns:
-The abstract requires revision. Authors should make their results more pronounced in the abstract.
We thank the reviews for this suggestion, we have revised and added several sentenced to the abstract to make it plump. The text goes as follow:
Massive amount of chalaza with near 400 metric tons was produced annually as waste in the liquid-egg industry. The present study aimed to look for utilizing chalaza as a natural emulsifier as natural emulsifiers for high internal phase emulsions (HIPEs) at the optimal production conditions to expand the utilization of such abundant material. To the author’s knowledge, for the first time, we report the usage of hen egg chalaza particles as particulate emulsifiers for Pickering (HIPEs) development. The chalaza particles with partial wettability were fabricated at different pH or ionic strengths by freeze-drying. The surface electricity of the chalaza particles was neutralized when the pH was adjusted to 4, where the chalaza contained a particle size around 1500 nm and held the best capability to stabilize the emulsions. Similarly, the chalaza reaches proper electrical charging (-6 mv) and size (700 nm) after the ionic strength was modified to 0.6M. Following the characterization of chalaza particles, we successfully generated stable Pickering HIPEs with up to 86% internal phase at proper particle concentrations (0.5–2%). The emulsion contained significant stability coalescence and flocculation during long term storage due to the electrical hindrance raised by the chalaza particles which absorbed on the oil-water interfaces. Different rheological models were tested on the formed HIPEs, indicating the outstanding stability of such emulsions. Concomitantly, a percolating 3D-networks was formed in the Pickering HIPES stabilized by chalaza which provided the emulsions with viscoelastic and self-standing features. Moreover, the current study provides an attractive strategy to convert liquid oils to viscoelastic soft solids without artificial trans fats.
-The rendition in lines 25-27: “Especially the protective effects of antioxidant egg-chalaza hydrolysates against chronic alcohol consumption-induced have been demonstrated liver steatosis in mice [4]”is poor. Revise and rewrite.
We thank the reviews for this advice, we have revised the sentence. The text goes as follow:
The protective effects of antioxidant egg-chalaza hydrolysates against chronic alcohol consumption-induced have been demonstrated in mice
-The statement in line 29 needs a reference.
We apologize for this obvious fault, we have added a reference on this sentence.
-Line 30 also has poor construction. Revise and rewrite.
We apologize for this, the sentence is revised and goes as follow:
At their outer ends, they merge with the outer thick white. Chalaziferous layer touching the vitelline membrane constructs the inner ends [5,6]
-Line 44: Is that a reference? If so, it should be according to the journal’s format.
We appreciate this suggestion, we have added the reference on this sentence:
-Lines 53-54 rendition is poor. Revise and rewrite.
We thank the reviewer for this idea, we have rewritten this paragraph. The text goes as follow:
These food biomass-based materials exhibits decent features to be used as Pickering emulsifiers to for the stabilization of HIPEs[29-34]. Moreover, the HIPEs stabilized by these particles have also been declared to be a promising template for the constructing of biocompatible scaffold for 3D cell culture
-The introductory section should provide more on state of the art of emulsions, types etc. These recent references are recommended: https://doi.org/10.3390/foods9020143; https://doi.org/10.3390/foods9050636; and https://doi.org/10.1016/j.lwt.2021.111024
Authors should try to improve the whole section too.
We thank the reviewer for these suggestions. We have added these three citations in the test. The overall quality of the whole section has also been improved.
-Authors must provide the location name in the materials/methods section. Local market, where?
We apologize for this mistake. The eggs were bought from tesco supermarket.
-Line 80: “…were used for...”
We apologize for this mistake. We have corrected this one.
-Line 147. You must provide reference(s).
We apologize for this mistake. We have corrected this one.
-Lines 149-155: The renditions are poor. Revise and rewrite.
We apologize for this fault. We have revised this sentence. The text goes as follow:
The protein analysis was corresponded to the former works[1,7], indicating that our separation method of chalaza was successful. The principal proteins are ovalbumin (54% of total protein), ovotransferrin (12%), ovomucoid (11%), and lysozyme (3.4%). The protein content of chalaza is similar to egg white. The two fastest migration proteins are α1-and α2 ovomucin, which are considered to be the carbohydrate-poor components of ovomucin. The slow migration one is carbohydrate-rich protein—β ovomucin. Overall, chalaza exhibits a very similar protein composition compared to egg white.
-What is the difference between sections 3.1 and 3.3 with respect to their titles?
We thank the reviewer for this question. To be more reader friendly, we have revised the title. The new titles are now:
3.1 Schematic models of chalaza
3.3 Particle size, ζ-potential and microstructure of chalaza particles
-Moreover, section 3.3 is poorly written and discussed, lacking flow and harmony. Authors failed to explain the importance of ionic concentration and pH levels in chalaza suspensions. Then, substantial references were lacking.
-Please, note that the whole of section 3.4 lack proper discussion, grammatical accuracy and adequate references!
We thank the reviewer for these questions, we have improved the quality of these two sections.
-Authors should ensure a constant flow between the abstract the main text.
-My final concern is that general grammatical errors and poorly written sentences could be found from the abstract to the acknowledgements. A CAREFUL revision is required.
We thank the reviewer for these questions, we have tried our best to improve quality of the manuscript.
Round 2
Reviewer 3 Report
The authors have greatly improved the manuscript. I recommend that it should be published.